# A Novel Hybrid Approach to the Diagnosis of Simultaneous Imbalance and Shaft Bowing Faults in a Jeffcott Rotor-Bearing System

Shyh-Chin Huang *, Sherina Octaviani and Mohammad Najibullah

Department of Mechanical Engineering, Ming Chi University of Technology, New Taipei City 24301, Taiwan; m11118035@mail2.mcut.edu.tw (S.O.); mohammad@mail.mcut.edu.tw (M.N.)
* Correspondence: schuang@mail.mcut.edu.tw

**Abstract:** Ensuring optimal performance and reliability in rotor-bearing systems is crucial for industrial applications. Imbalances and shaft bowing in these systems can lead to decreased efficiency and increased vibrations. The early detection and mitigation of a rotor's faults are essential, and model-based fault identification has gained much attention in the manufacturing industry over the years. Over the past two decades, however, the development of fault diagnosis rules with data-driven and artificial intelligence (AI) methods has become a trend, and in the foreseeable future the combination of AI with big data will become mainstream. Nevertheless, the critical role of rotating machinery in manufacturing introduces a challenge, as often insufficient fault data are available. This limitation renders the establishment of diagnostic rules using data-driven methods and AI technologies impractical. In light of these challenges, this study proposes a novel hybrid approach that combines a physical model with machine learning (ML) techniques for the diagnosis of multi-faults (imbalances and shaft bowing are demonstrated) in a Jeffcott rotor. To overcome the lack of real-world labeled fault datasets, a physics-based Jeffcott rotor model is first derived and then used to generate abundant fault datasets for ML. Subsequently, simulated data are employed for the training of an artificial neural network (ANN), enabling the network to learn from and analyze the vast array of generated data. The results prove that a well-trained feed-forward neural network (FNN) can accurately isolate and diagnose imbalance and shaft bowing faults using the simulated and real data from the Jeffcott rotor experiment. These physics-based and ML approaches prove effective particularly for multi-faults, offering new possibilities for advanced rotor system monitoring and maintenance strategies in industrial applications.

**Keywords:** hybrid approach; multi-fault diagnosis; machine learning; ANN; imbalance and bow

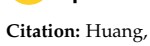



## 1. Introduction

Rotating machines find widespread utilization across diverse domains, including in the manufacturing sectors, wind energy generation, aviation propulsion systems, electric motor assemblies, marine propulsion systems, and mining equipment, owing to their notable attributes of elevated efficacy and robustness. Within the framework of Industry 5.0, a significant transformation can be observed, whereby the convergence of cutting-edge technologies and the promotion of sustainable practices in the manufacturing sector are emphasized. In this context, the utilization of rotating machines assumes a crucial and prominent role. Advancements in cutting-edge technologies in the field of material sciences have led to the development of rotating machines that are both faster and lighter, while also being capable of withstanding prolonged periods of operation. Numerous operational irregularities, including but not limited to mass imbalances, bowed shafts, cracked shafts, misalignment, elevated temperatures, quick acceleration, and frequent load fluctuations, all contribute to reduced efficiency and premature mechanical breakdown. Consequently, these issues result in unplanned periods of inactivity and financial detriment. Hence,

ensuring the efficient maintenance and accurate fault detection and diagnosis (FDD) of rotating machines holds significance in achieving the objectives of Industry 5.0, whereby the convergence of efficiency, sustainability, and productivity is emphasized [1].

Vibration fluctuations, which are highly sensitive to both minor structural variations and changes in the operational process, are the basis for condition monitoring and diagnostics in rotating equipment. Any fault that occurs in a rotor will alter its vibrational behavior, and the degree of this impact depends largely on the various fault types. Given this, vibration-based diagnostics (VBD) has gained popularity and is often used in practice to diagnose a variety of rotor faults [2]. Walker et al. [3] reviewed the recent advances in the VBD and prognosis of rotors with eight common problems. The approaches for fault diagnosis can be broadly divided into three categories: (1) physics-based or model-based approaches; (2) data-driven approaches; (3) the combination of various approaches, called the hybrid method.

Data-driven approaches to rotor diagnosis utilize statistical theories or artificial intelligence (AI) algorithms to process the collected data, especially fault data, for fault diagnosis [4]. In recent years, data-driven approaches have become promising with the advancement of signal processing techniques such as empirical mode decomposition (EMD), Hilbert–Huang transforms (HHT) [5–7], and wavelet transforms [8–10]. In the foreseeable future, data-driven methodologies combined with artificial intelligence (AI) technology will remain well-liked and fruitful. Given the progress in the field of vibration analyses, there is increasing interest in the application of AI techniques for fault diagnosis in rotating machinery. The integration of AI approaches has gained significant recognition and traction in both academic and industrial circles, presenting a promising avenue for addressing industrial challenges. Walker et al. [11] proposed automating the localization of imbalanced faults using ANN techniques. Mohamed et al. [12] proposed a method for diagnosing faults in rotating machinery using a frequency domain vibration analysis and neural network (NN) pattern classification. A variety of unbalanced fault types have been localized with high precision on dynamic rotor test rigs using ANNs, providing benefits by reducing the number of sensors used. Liu et al. [13] provided a comprehensive analysis of the AI-based research and development for rotating machinery fault diagnostics, encompassing both theoretical and practical standpoints. Aneesh et al. [14] recently presented a comprehensive review of the role of AI in rotor fault diagnosis. They discussed the difference between traditional machine learning (ML) and deep learning approaches.

For the success of a data-driven approach, a sufficient amount of accurate data must be available for network training. However, industrial plant rotor systems always have insufficient datasets. In that regard, model-based techniques designed for fault identification originating from physical models of rotor systems (complete or partial) have been offered for years. Using model-based techniques, numerous studies have detected rotor faults with success. Edward et al. [15] used a model-based identification approach in the frequency domain to identify an imbalance on a test rig. Bachschmid and Pennacchi [16] used an experimental validation approach to support model-based methods for fault location, severity assessment, and fault classification. Sekhar [17] applied this methodology to concurrently identify instances of imbalances and cracking within a rotor-bearing system. Jain and Kundra [18] employed model-based techniques to detect imbalances and cracks and experimentally validated their imbalance identification results conducted on a test rig. Sinha et al. [19] estimated the imbalance and misalignment of a flexible rotating machine based on a single run-down procedure. Lees et al. [2] provided a thorough analysis of a model-based rotating machine identification system. They discussed various methods for deriving foundation models from operational data and subsequently updating them. Jalan and Mohanty used an experimental model-based method to identify imbalance and misalignment issues in a rotor-bearing system [20]. However, there are numerous applications for this technique, such as in the studies conducted by one particular work group [21–23], who have made significant contributions to the field of rotor fault identification using a model-based method. In a very recent study, Lin et al. [24] derived a novel model-based

approach in which model parameters such as bearing constants and initial imbalances are identified in the first phase and progressive imbalances based on daily operational data are used to identify them in the second phase. The imbalance can, therefore, be monitored online and in real time. In general, the model-based method yields the most accurate results, which are consistent with the physical theory; however, model derivation and verification processes are extremely time-consuming and must be performed by rotor specialists.

Numerous ongoing studies on imbalances in rotating machinery have been performed, spanning a wide range of topics, such as imbalance misalignment [25], imbalance diagnostics [26], modeling methodologies to help with imbalance prediction, and lab-based experimental investigations. There are a number of comprehensive literature studies that outline the range of imbalance prediction research, including those by Edward et al. [27], Randall [28], and Walker et al. [29].

Shaft bowing refers to a deviation from the perfect alignment of the geometrical axis of the rotor shaft. This phenomenon is often attributed to factors such as heat gradients experienced during the initiation and cessation of operation in thermal turbo turbines, material creep, manufacturing discrepancies, and other contributing factors. Depending on the amount and location of the bend, shaft bowing can cause an excessive amount of vibration in a machine [30]. Numerous scholarly studies have been conducted to discern the impacts of shaft bowing in rotating machinery, assess their dynamic properties, and implement suitable corrective measures [31]. The impact of residual rotor bowing on rotor vibrations was examined in a seminal study conducted by Nicholas et al. [32,33].

Researchers have studied how residual shaft bowing affects the imbalance response in a simplified rotor model. They examined different combinations of bowing and imbalances to understand their interaction. Additionally, they proposed three distinct balancing methods based on their findings. Flack et al. [34] employed a transfer matrix technique to forecast the imbalance response of a Jeffcott rotor subjected to bowing. The rotor was affixed to various fluid film bearings, and the outcomes were subsequently contrasted with prior experimental investigations. Shiau et al. [35] investigated the impact of residual shaft bowing on the dynamic response of a simply supported single disk rotor. Their study systematically examined the interplay of factors such as disk skewing, mass imbalances, and the positioning of the disk between the bearings. Rao et al. [36] scrutinized a warped Jeffcott rotor model across diverse bowing scenarios, revealing instances of self-balancing and phase jumping. Srinivas et al. [37] employed an ANN and wavelet transform to categorize a rotor system that experienced both shaft bowing and an imbalance. This was achieved by monitoring the vibrations in the transverse and axial directions. Rezazadeh et al. [38] recently presented research where they used a combination of a WTS (wavelet transform spectrum) with LSTM (long short-term memory) and an SVM (support vector machine) to detect imbalances and shaft bowing in rotor systems. When there are multiple faults, the vibrations caused by these faults are coupled with each other in a linear or nonlinear manner. As a result, the observed vibration signals are quite complicated, making it challenging to detect each fault using traditional methods.

Even though model-based approaches have been used to diagnose and anticipate rotor defects for many years, there are drawbacks, including the need for lengthy models and a lack of self-adaptability to aging machinery. Therefore, with the aid of modern signal processing, hybrid methods that make use of AI techniques and physical modeling have appeared. Regarding the commonly named hybrid approaches, in light of their combination types, they can be divided into three categories—serial combinations (method fusion) [39], parallel combinations (output fusion) [40], and mixed combinations. Physical model may be derived and employed along with these hybrid methods, although mainly just to generate the residual in the parity equation. The present research, however, proposes a novel combination of physics-based and ML techniques, in which a physical model generates synthetic data for ML training. An exemplar in this domain is the work by Huang et al. [41,42], who adeptly introduced the concept of a hybrid approach by incorporating a neural network (NN) alongside physical modeling for imbalance identification. Djeziri

et al. [43] used a hybrid method for fault prognosis based on a physical model, data clustering, and the geolocation principle to predict the remaining useful life (RUL) for wind turbine systems. Recently, Wilhelm et al. [44] reviewed hybrid approaches for FDD that combine data-driven analyses with physics-based and knowledge-based models to overcome a lack of data and increase the FDD accuracy. Moreover, Fang et al. [45] presented a fault diagnosis and prognosis method based on a hybrid approach that combines structural and data-driven techniques.

The objective of this study is to create a hybrid approach that is applicable for the diagnosis of multiple faults in rotor-bearing systems. The essence of this innovative method is as follows:

1. We will establish a physical and mathematical model of a Jeffcott rotor-bearing system containing simultaneous imbalance and shaft bowing faults and identify the model parameters, including the imbalance and shaft bowing characteristics, from calculations and experimental tests.

2. After parameter identification, the physical model will be used to generate sufficient sets of simulated data for ANN-supervised training, which will help to produce a more reliable model. A trained ANN can be integrated into a Jeffcott rotor monitoring system for the online diagnosis of imbalance and shaft bowing fault components using simulated and experimental data from Jeffcott rotor experiments.

The remainder of this paper's structure is as follows. In Section 2, the physical model of a Jeffcott rotor subjected to an imbalance and shaft bowing is described. Section 3 introduces the hybrid methodology employed in the present study. The numerical analysis and experimental verification process are described in Section 4. Finally, in Section 5, our concluding remarks are presented to justify the effectiveness of this approach.

## 2. Physical Model of a Jeffcott Rotor with Simultaneous Imbalance and Shaft Bowing Faults

The schematic design involves a purely physics-based approach. This design, as elucidated in Figure 1a, provides a visual representation of the Jeffcott rotor systems consisting of an imbalance and shaft bowing with a rigid disk, supported by two simple bearings, while Figure 1b shows the disk's geometric relations. It is assumed that the shaft-bearing component possesses only stiffness $K$ and light damping $B$. Here, $O$ stands for the center of the rotation axis, $C$ stands for the disk's geometric center, $e$ is the distance of the rotor's imbalance from the geometric center, $\Omega$ represents the rotational speed, $s$ represents the residual shaft bowing, $m$ represents the imbalance mass, $M$ represents the disk mass, $\theta$ represents the shaft bowing angle, and $\alpha$ represents the imbalance angle, both of which reference a key phasor (KP).

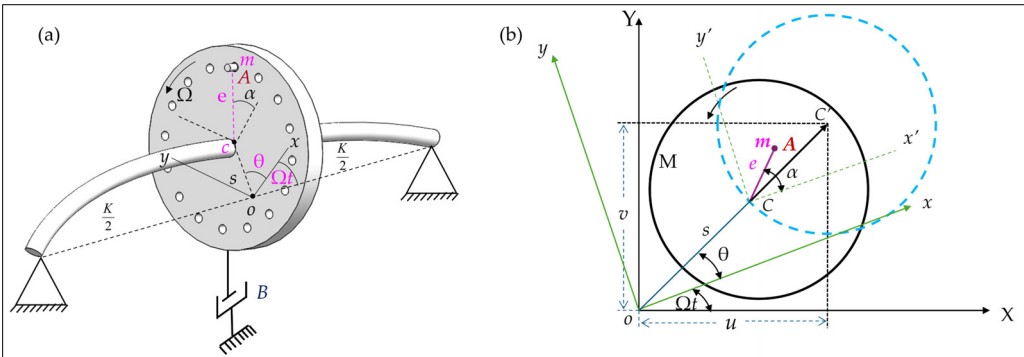

**Figure 1.** Schematic diagram of a Jeffcott rotor (**a**) with simultaneous imbalance and shaft bowing faults and (**b**) the disk's geometric relations.

The rotor imbalance is commonly specified by a term of $U = me$, with a unit of g·mm or kg·m. The acceleration of the imbalance mass at point $A$ can be calculated based on fundamental dynamics, i.e.:

$$\mathbf{a}_A = \ddot{u}\mathbf{i} + \ddot{v}\mathbf{j} - \Omega^2[e \cdot \cos(\Omega t + \alpha)\mathbf{i} + e \cdot \sin(\Omega t + \alpha)\mathbf{j}] \tag{1}$$

where $\mathbf{i}$ and $\mathbf{j}$ respectively denote the unit vector in the X and Y directions. The inertia force resulting from the imbalanced mass $m$ can be represented by:

$$-m\mathbf{a}_A = [-m\ddot{u} + me\Omega^2(\cos\Omega t + \alpha)]\mathbf{i} + [-m\ddot{v} + me\Omega^2(\sin\Omega t + \alpha)]\mathbf{j} \tag{2}$$

The equations of motion (EOMs) for the Jeffcott rotor with simultaneous imbalance and shaft bowing faults can be obtained by using Newton's second law:

$$\Sigma F_u = M\ddot{u} \tag{3}$$

$$\Sigma F_v = M\ddot{v} \tag{4}$$

Subsequently, the EOMs subject to the imbalance and shaft bowing excitation in the X and Y directions are summarized as:

$$(M+m)\ddot{u} + B_x\dot{u} + K_x u = K_x s \cos(\Omega t + \theta) + me\Omega^2 \cos(\Omega t + \alpha) \tag{5}$$

$$(M+m)\ddot{v} + B_y\dot{v} + K_y v = K_y s \sin(\Omega t + \theta) + me\Omega^2 \sin(\Omega t + \alpha) \tag{6}$$

or in terms of nominal vibration forms as:

$$\ddot{u} + 2\zeta_x\omega_{nx}\dot{u} + \omega_{nx}^2 u = s\omega_{nx}^2 \cos(\Omega t + \theta) + \frac{U}{M_T}\Omega^2 \cos(\Omega t + \alpha) \tag{7}$$

$$\ddot{v} + 2\zeta_y\omega_{ny}\dot{v} + \omega_{ny}^2 v = s\omega_{ny}^2 \sin(\Omega t + \theta) + \frac{U}{M_T}\Omega^2 \sin(\Omega t + \alpha) \tag{8}$$

where $M$, $B$, and $K$ respectively represent the mass, damping, and stiffness. Here, $M_T = M + m \approx M$ denotes the total mass of the rotor, while $\omega_{ni} = \sqrt{K_i/M_T}$ and $\zeta_i = C_i/2\sqrt{K_i M_T}$, $i = x, y$ represent the system's natural frequency and damping ratio in the X and Y directions, respectively.

The right-hand terms in Equations (7) and (8) describe the excitation forces caused by shaft bowing and imbalances, respectively. The vibrational responses of $u$ and $v$ due to shaft bowing and imbalances can be separately solved and superimposed for the total responses. Nonetheless, identifying the rotor's imbalance ($U,\alpha$) and shaft bowing ($s,\theta$) from the total responses is a backward process involving nonlinear functions of $\alpha$ and $\theta$. Therefore, nonunique solutions exist, which causes difficulty in almost all cases of multi-fault diagnosis.

Let us first provide the responses to these two types of faults separately and then introduce the idea of combining the physical modeling and machining learning techniques for multi-fault diagnosis. The EOM of $u$ due to bowing alone is:

$$\ddot{u}_b + 2\zeta_x\omega_{nx}\dot{u}_b + \omega_{nx}^2 u_b = s\omega_{nx}^2 \cos(\Omega t + \theta) \tag{9}$$

where the subscript $b$ represents the term "bowing". The solution of Equation (9) can be readily obtained from any standard textbook on vibrations as:

$$u_b = s \cdot A_x \cos(\Omega t + \theta - \lambda_x) \tag{10}$$

where $A_x$ is the amplification factor, which can be expressed as follows:

$$A_x = \frac{1}{\sqrt{(1 - \tau_x^2)^2 + (2\zeta_x\tau_x)^2}} \tag{11}$$

$$\tau_x = \Omega/\omega_{nx} \tag{12}$$

$$\lambda_x = \tan^{-1} \frac{2\zeta_x \tau_x}{1 - \tau_x^2} \tag{13}$$

Here, $\tau$ is the speed ratio or frequency ratio and $\lambda$ is the response phase lag.

In most cases, $m<<M$; thus, $M_T \approx M$. Similarly, the EOM caused by the imbalance can be written as:

$$\ddot{u}_u + 2\zeta_x \omega_{nx} \dot{u}_u + \omega_{nx}^2 u_u = \frac{U}{M} \Omega^2 \cos(\Omega t + \alpha) \tag{14}$$

The subscript $u$ represents a state of imbalance. Similarly, the response due to the imbalance can be solved as:

$$u_u = \frac{U \cdot \tau_x^2}{M} A_x \cos(\Omega t + \alpha - \lambda_x) \tag{15}$$

Through superposition, the responses in the X and Y directions can be expressed as follows:

$$u(t) = A_x \big[ \frac{U\tau_x^2}{M} \cos(\Omega t + \alpha - \lambda_x) + s \cos(\Omega t + \theta - \lambda_x) \big] \tag{16}$$

$$v(t) = A_y \big[ \frac{U\tau_y^2}{M} \sin(\Omega t + \alpha - \lambda_y) + s \sin(\Omega t + \theta - \lambda_y) \big] \tag{17}$$

The responses of Equations (16) and (17) can be further rearranged in terms of the $\cos(\Omega t)$ and $\sin(\Omega t)$ components as follows:

$$u(t) = f_1 \cos(\Omega t) + f_2 \sin(\Omega t) \tag{18}$$

$$v(t) = f_3 \cos(\Omega t) + f_4 \sin(\Omega t) \tag{19}$$

where $f_1, f_2, f_3,$ and $f_4$ are the four features as functions of the four fault variables $U, \alpha, s,$ and $\theta$, respectively, defined as:

$$f_1 = A_x \big[ \frac{U\tau_x^2}{M} \cos(\alpha - \lambda_x) + s \cos(\theta - \lambda_x) \big] \tag{20}$$

$$f_2 = -A_x \big[ \frac{U\tau_x^2}{M} \sin(\alpha - \lambda_x) + s \sin(\theta - \lambda_x) \big] \tag{21}$$

$$f_3 = A_y \big[ \frac{U\tau_y^2}{M} \sin(\alpha - \lambda_y) + s \sin(\theta - \lambda_y) \big] \tag{22}$$

$$f_4 = A_y \big[ \frac{U\tau_y^2}{M} \cos(\alpha - \lambda_y) + s \cos(\theta - \lambda_y) \big] \tag{23}$$

or in terms of the feature vector as:

$$\mathbf{f}_{4\times 1} = \{f_1, f_2, f_3, f_4\}^T \tag{24}$$

The fault variables associated with imbalances and shaft bowing can be written as a fault vector:

$$\mathbf{m}_{4\times 1} = \{U, \alpha, s, \theta\}^T \tag{25}$$

The subject of the diagnosis is to identify Equation (25) from the data from Equation (24) using calculations or measurements. It is apparent that due to the variability and nonlinearity, the solution is nonunique. The ANN provides a robust alternative to solve this type of problem.

Note that the features shown in (20)–(23) are not only functions of the fault variables but also functions of system variables such as the speed ratio $\tau$ and damping ratio $\lambda$. The impact of the damping and speed ratios on the diagnosis accuracy for imbalances and shaft bowing will be discussed in Section 4.

## 3. Hybrid Methodology

In this paper, the authors propose a comprehensive approach for establishing a real-time imbalance and shaft bowing diagnosis system for a rotor system and use a Jeffcott rotor-bearing system as the application example. This approach combines mathematical modeling techniques with ML-based prediction methods to enable the accurate and timely detection of multiple faults in the system, as shown in Figure 2. The ability to assess the onset of faults is a crucial aspect of both physics-based and machine learning approaches when analyzing the overall system.

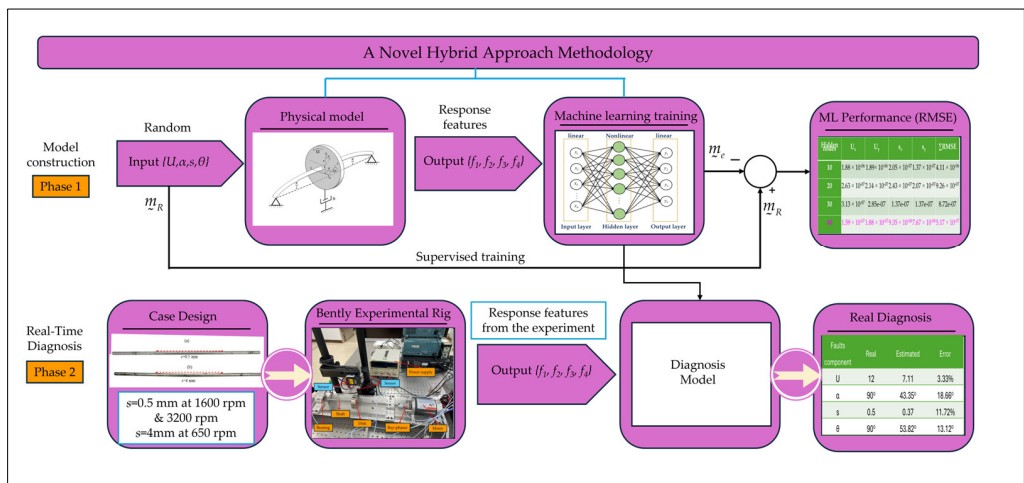

**Figure 2.** A novel hybrid methodology for real-time imbalance and shaft bow diagnoses.

ML techniques are increasingly being used to diagnose faults in rotating machinery by analyzing data from sensors on the machinery. These models can learn to detect patterns that indicate faults and classify the type and severity of the problem, which can help maintenance teams diagnose issues earlier and prevent costly downtime. To overcome this challenge, a hybrid approach will be developed. This approach will combine machine learning techniques with expert knowledge and physical models to augment the limited fault data.

Data-driven methodologies leverage statistical theories and artificial intelligence (AI) algorithms to analyze gathered data—particularly fault data—in order to facilitate fault diagnosis. However, the efficacy of data-driven techniques is contingent upon a substantial amount of accurately labeled data being available for network training. Notably, researchers frequently encounter challenges in acquiring sufficient labelled datasets for industrial rotor systems. To address this data constraint issue, the physical model employs a knowledge-based approach, presenting a viable way to overcome the lack of relevant data.

Nevertheless, it is imperative to acknowledge the limitations of this data-driven approach, particularly in scenarios involving multiple faults, where its reliability may be compromised. In such instances, utilizing a physical model to generate fault data for ANN training emerges as a more dependable solution, enhancing the robustness of the overall diagnostic process.

Two steps are involved in this process, namely model construction and real-time diagnosis, as shown in Figure 2. The model construction phase involves a combination of physical modeling and ML techniques. By using a physical model, one can generate sufficient sets of simulated data for ANN-supervised training, which helps to produce a more reliable model for diagnosing imbalances and shaft bowing in a Jeffcott rotor-bearing system. The simulated dataset is randomly generated using a systematic approach, with the imbalance parameters ranging from 0 to 1000 (g·mm) and the bowing parameters ranging from 0 to 500 (mm). Next, the derived model is implemented into a real rotor system for real-time diagnosis, as displayed in phase 2. In this phase, cases with two

different residual bowing faults plus various imbalance combinations are tested to acquire the real response features from an experimental setup involving a rotor rig. Note that in the designed scenarios the bowing only occurs under two conditions, $s = 0.5$ mm run at 1600 rpm and 3200 rpm and $s = 4$ mm run at 680 rpm, due to the difficulty of bending the shaft to achieve a small, permanent bow. Moreover, in the $s = 4$ mm bowing case, the rotor can be run only at very low speed due to safety concerns.

The real response features *($f_1$, $f_2$, $f_3$, and $f_4$)* are then calculated from the measured response of the rotor rig and fed into the trained FNN for instant fault component identification, i.e., $\mathbf{m}^T = \{U, \alpha, s, \theta\}$. By combining phase 1 and phase 2, this hybrid approach can be used to overcome the data availability limitations and provide an accurate diagnosis model for imbalances and bow faults in rotating machinery.

To achieve this goal, we employed the most common ML-based approaches used for rotor fault diagnosis, which are ANNs. ANNs can learn to classify different types of faults based on vibrations, currents, or acoustic data from sensors on the machinery. In particular, ANNs can be utilized to diagnose imbalances and shaft bowing in rotating machinery, which are common problems that can lead to excessive vibrations, reduced machine performance, and even catastrophic failure.

Among the various types of ANNs, we had tested various types such as feed-forward neural networks (FNNs), RNNs, and LSTMs and found that their differences in accuracy are almost nondifferentiable. Hence, an FNN was selected for its simple structure and computational efficiency. In the FNN, the information flows in one direction, from the input layer through one or more hidden layers to the output layer. By training the FNN with historical data from the rotor system, it can learn the underlying patterns and relationships between the various system parameters and multiple faults. This enables the FNN to make predictions about future fault events based on real-time sensor data. Figure 3 demonstrates the applications of the machine learning framework for diagnosing multiple faults in a Jeffcott rotor-bearing machine, showcasing its potential for multifaceted fault detection and analyses.

1.  Acquiring data: The physical model is used to generate the datasets randomly. Nevertheless, the training set can be generated from measured data as much as one need. It is notable that these imbalance and shaft bowing components, $\mathbf{m}^T = \{U, \alpha, s, \theta\}$, are random inputs to the physical model, and the response features components at the disk centre after the forwarding calculation are the output, i.e., Equation (24).

2.  Data preparation: The generated datasets will be considered raw data for further processing via supervised training. The simulated inputs and outputs are reversed during the network training. The response components $\mathbf{f}^T$, i.e., the 4 parameters, are used as the input to the ANN, and the 4 parameters $\mathbf{m}^T$ are used for the target.

3.  ANN architecture: An appropriate model needs to be selected and configured for the problem at hand. Typically, a feed-forward neural network (FNN) with one or more hidden layers is used. There are several parameters in each model that can be varied to arrive at the best model parameter for each case. The proposed framework is modeled using MATLAB version: 9.15.0 (R2023b) software.

4.  Model training: The FNN is trained with a single hidden layer architecture with a varying number of nodes using 10,000 datasets, which are randomly generated from a physical model. The first 70 percent of the datasets are utilized for training, the second 15 percent are used for validation, and the final 15 percent are used for testing.

5.  Model testing: To rigorously evaluate the FNN model's performance, the root mean squared error (RMSE) is employed to evaluate the alignment between the randomly generated values (so-called real data) and estimated values. A lower RMSE value signifies better accuracy and model performance. The formula for the RMSE is:

$$RMSE = \sqrt{\frac{1}{n}\sum_{i=1}^{n}(y_i - \hat{y}_i)^2} \qquad (26)$$

where $y_i$ denotes the real value for the *i*th point, $\hat{y}_i$ denotes the estimated value, and n denotes the number of datapoints.

6.  Diagnosis: The trained FNN is tested using simulated and real data acquired from the experimental Jeffcott rotor setup to diagnose the multi-fault components, i.e., $U$, $\alpha$, $s$, and $\theta$. Based on the output of the FNN, the machine operator can monitor the growth of imbalances and shaft bowing faults to take necessary actions.

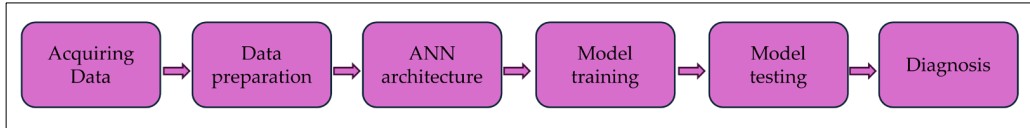

**Figure 3.** ML approach to diagnosing multiple faults in a Jeffcott rotor-bearing system.

## 4. Numerical Analysis and Experimental Verification

### 4.1. Numerical Analysis Using Simulated Data

The detection of imbalances and shaft bowing in Jeffcott rotor systems is an immensely significant task within numerous industrial domains. To tackle this challenge, the use of ANN models has proven to be highly effective. Our investigation commenced by employing an FNN that comprised multiple hidden layers, an input layer, and an output layer, all lacking feedback connections or loops [46].

In the context of diagnosing multiple faults, the input layer adeptly receives the vibration signals or response components originating from the horizontal and vertical directions of the Jeffcott rotor system. Subsequently, these signals are skillfully processed by the neurons residing within the concealed layers, which judiciously employ activation functions such as the tan-sigmoid function or the rectified linear unit (ReLU) function. The outcome obtained from the hidden layer is then skillfully mapped to the ultimate output through the neurons residing within the output layer. It is worth noting that the output layer typically employs a linear transfer function, such as the purelin transfer function, to carry out this mapping process.

To train the FNN, 10,000 datasets are randomly generated, and 70% of them are used for training, 15% for validation, and the last 15% for testing. The Levenberg–Marquardt backpropagation (LM) training algorithm, specifically the *trainlm* training function in MATLAB, was employed to train the network. To determine the most suitable number of hidden nodes, a trial-and-error approach was employed, as there is no precise method that can be used to select this parameter based solely on the number of inputs and outputs. However, in many ANN applications, including for FNNs applied to rotor systems, a single hidden layer has been proven sufficient [37,47]. Once the FNN is trained, it can be used to accurately detect and diagnose imbalances and shaft bowing in Jeffcott rotor systems based on the input vibration signals.

To examine the performance of the models, networks were trained with 10, 20, 30, 40, 50, and 60 hidden nodes. The results of the study revealed that a single hidden layer with 40 nodes had the lowest RMSE, as shown in Figure 4. The number of errors drops rapidly and reaches the minimum for the configuration involving 40 nodes. The elapsed time periods vary with the node number, as also shown in Figure 4, and it can be seen that the time increases almost linearly with the number of nodes. The errors of the testing set were reported in Table 1 in terms of ($U_x$, $U_y$, $s_x$, $s_y$) rather than ($U$, $s$, $\alpha$, $\theta$), because the phase angle values ($\alpha$ and $\theta$) did not align accurately because of their periodic nature. This change was made to mitigate concerns regarding potentially misleading outcomes. In Table 1, the lowest numbers of errors for the four components all occur with 40 nodes.

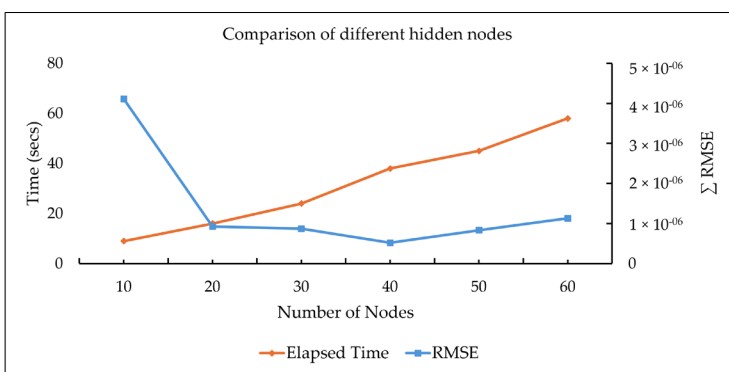

**Figure 4.** Comparison of different hidden node setups.

**Table 1.** RMSE values for the imbalance and shaft bowing components of different node setups.

| Hidden Nodes | Elapsed Time (s) | $U_x$ | $U_y$ | $s_x$ | $s_y$ | $\sum$RMSE |
|---|---|---|---|---|---|---|
| 10 | 9 | $1.88 \times 10^{-6}$ | $1.89 \times 10^{-6}$ | $2.05 \times 10^{-7}$ | $1.37 \times 10^{-7}$ | $4.11 \times 10^{-6}$ |
| 20 | 16 | $2.63 \times 10^{-7}$ | $2.14 \times 10^{-7}$ | $2.43 \times 10^{-7}$ | $2.07 \times 10^{-7}$ | $9.26 \times 10^{-7}$ |
| 30 | 24 | $3.13 \times 10^{-7}$ | $2.85 \times 10^{-7}$ | $1.37 \times 10^{-7}$ | $1.37 \times 10^{-7}$ | $8.72 \times 10^{-7}$ |
| 40 | 38 | $1.59 \times 10^{-7}$ | $1.88 \times 10^{-7}$ | $9.35 \times 10^{-8}$ | $7.67 \times 10^{-8}$ | $5.17 \times 10^{-7}$ |
| 50 | 45 | $2.18 \times 10^{-7}$ | $2.73 \times 10^{-7}$ | $1.57 \times 10^{-7}$ | $1.87 \times 10^{-7}$ | $8.34 \times 10^{-7}$ |
| 60 | 58 | $4.59 \times 10^{-7}$ | $4.34 \times 10^{-7}$ | $1.17 \times 10^{-7}$ | $1.18 \times 10^{-7}$ | $1.13 \times 10^{-6}$ |

During our investigation of the performance of the FNN using simulated data, we proposed a variety of fault conditions so that the diagnosis accuracy rates could be observed and compared. Different fault combinations can be classified as imbalance-dominant, shaft-bowing-dominant, and equal. When the imbalance dominates, the $U$ values range from 0.6 to 0.9 kg·m and the $s$ values from 0.5 to 0.1 mm. This means that the imbalance component is 100 times the shaft bowing component. In contrast, shaft bowing dominance involves $U$ values ranging from 0.00001 to 0.00002 kg·m and $s$ values ranging from 2 to 3 mm, opposing the imbalance scenario. The $U$ and $s$ parameters vary from 0.002 to 0.003 kg·m and 2 to 3 mm, respectively, in the equal case.

In addition, the effects of imbalances and shaft bowing vary with the rotational speed. The FNN performance results under three different operational ranges, namely sub-critical speed ($\tau < 1$), near-critical speed ($\tau \approx 1$), and trans-critical speed ($\tau > 1$), were also investigated. The RMSE was used as the metric for the performance evaluation in each scenario, and lower RMSE values indicate better performance or closer alignment with the FNN predictions.

The data illustrated in Table 2 demonstrate the performance of the FNN across various fault dominations at different speeds. Let us first look at the third column for the imbalance-dominant case, where it can be observed that the estimated error sum of the FNN is the lowest (green face) when running at near-critical speed $\tau \approx 1$, then $\tau > 1$ and $\tau < 1$. The identification error sum of the shaft bowing case, however, exhibits no significant differences for the different speed ranges. For the cases that are shaft-bowing-dominant, there are no significant differences in imbalance and bowing estimation errors under various speeds. When these two faults are of equal weight, the lowest estimation errors for imbalances and shaft bowing happen in different speed regions. Imbalances are best identified at $\tau > 1$, while the bowing identification process shows the lowest error with the rotor running at $\tau < 1$.

**Table 2.** Comparison of RMSE values for fault components under different speed ratios.

| | Fault Components | Imbalance-Dominant | Shaft-Bowing-Dominant | Equal |
|---|---|---|---|---|
| $\tau < 1$ | | | | |
| | $U_x$ | 0.0323 | 0.3101 | $4.69 \times 10^{-5}$ |
| | $U_y$ | 0.0581 | 0.3140 | $6.01 \times 10^{-5}$ |
| | $s_x$ | 0.2924 | 0.0147 | $1.37 \times 10^{-5}$ |
| | $s_y$ | 0.3398 | 0.0215 | $1.66 \times 10^{-5}$ |
| $\tau \approx 1$ | | | | |
| | $U_x$ | 0.0200 | 0.2937 | $3.96 \times 10^{-6}$ |
| | $U_y$ | 0.0200 | 0.3295 | $6.87 \times 10^{-6}$ |
| | $s_x$ | 0.3031 | 0.0238 | $3.31 \times 10^{-5}$ |
| | $s_y$ | 0.2727 | 0.0146 | $6.67 \times 10^{-5}$ |
| $\tau > 1$ | | | | |
| | $U_x$ | 0.0288 | 0.3557 | $6.59 \times 10^{-6}$ |
| | $U_y$ | 0.0315 | 0.3008 | $3.11 \times 10^{-6}$ |
| | $s_x$ | 0.3459 | 0.0105 | $4.07 \times 10^{-5}$ |
| | $s_y$ | 0.2873 | 0.0221 | $6.99 \times 10^{-5}$ |

From the results shown above, it can be concluded that imbalances are better diagnosed at higher speeds ($\tau \approx 1$ or $\tau > 1$), while the shaft bowing is better diagnosed at lower speeds ($\tau < 1$). This conclusion can be explained and verified using the frequency responses of these two types of faults, as shown in Figure 5. In this figure, it can clearly be seen that the imbalance response is significantly magnified at higher speeds, while conversely the shaft bowing response diminishes with increasing speeds. In other words, shaft bowing dominates at sub-critical speeds and imbalances take over after the critical speed is reached. Both reach their maximum response at near-critical speeds.

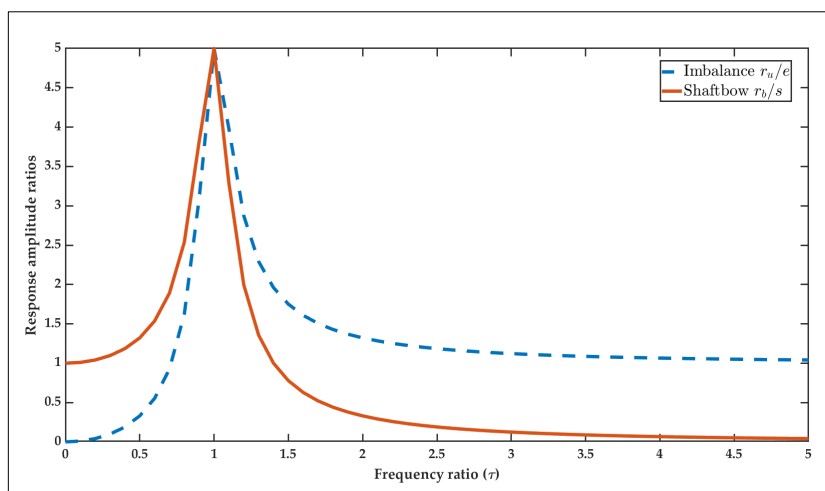

**Figure 5.** Frequency responses for imbalance and shaft bowing faults.

Moreover, in Table 2, it is noteworthy that the equal scenario case exhibits much smaller RMSE values than the others for both the imbalance and shaft bowing faults. This means that the FNN diagnosis of the simultaneous imbalance and shaft bowing faults would be the most accurate when the shaft bowing and imbalance faults are not unduly dominant.

Damping is another factor that affects the response amplitude and phase. In most rotor-bearing systems, the damping ratio is relatively low. Nevertheless, to look into the damping effect in depth, cases with equal fault–weight ratios but with different damping ratios (underdamped, critically damped, and overdamped) were tested at sub-critical speeds ($\tau < 1$). The diagnosis accuracy results for the fault components are shown in Table 3 as RMSE values.

**Table 3.** Comparison of RMSE values for fault components under different damping ratios.

| Fault Components | Average RMSE | | |
|:---:|:---:|:---:|:---:|
| | $\zeta < 1$ | $\zeta = 1$ | $\zeta > 1$ |
| $U_x$ | $1.17 \times 10^{-5}$ | $2.93 \times 10^{-5}$ | $1.05 \times 10^{-5}$ |
| $U_y$ | $1.35 \times 10^{-5}$ | $2.43 \times 10^{-5}$ | $1.62 \times 10^{-5}$ |
| $s_x$ | $8.75 \times 10^{-5}$ | $8.21 \times 10^{-5}$ | $8.66 \times 10^{-5}$ |
| $s_y$ | $6.81 \times 10^{-5}$ | $9.15 \times 10^{-5}$ | $7.27 \times 10^{-5}$ |

As seen in Table 3, the RMSEs all fall in the order of $10^{-5}$ and the effects of the damping on the accuracy are not significant.

### 4.2. Experimental Verification and Real-Time Diagnosis

The model RK 4 Bently Nevada experimental rig depicted in Figure 6 was used for the rotor fault experiment. This consisted of a motor attached to a single-disk rotor. The motor ran counterclockwise as viewed from the motor side. The rotor shaft was upheld by simple, identical bearings of an unknown stiffness. The diameter of the rotor shaft was 10 mm. A disk of 75 mm in diameter and 800 g in mass was mounted on the rotor shaft by radial screws. There were 16 tapped holes symmetrically placed on each side of the disk with flat faces at $e$ = 30 mm to attach any desired amount of imbalance mass. Memstec Glory Laser CD3S-30 and CD3S-50 sensors were precisely positioned on the corresponding disk along the X and Y axes, which were positioned at a separation angle of 90 degrees. After installation, the sensors were integrated into the power supply system to ensure a reliable and uninterrupted power source to ease the measurement operation.

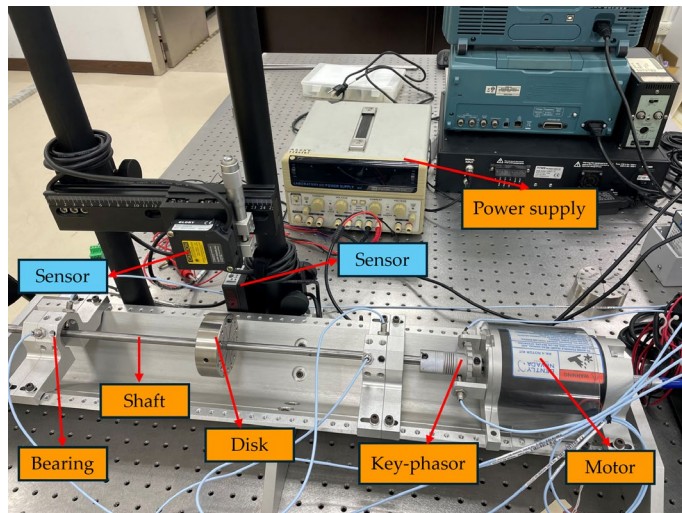

**Figure 6.** The rotor experiment platform.

The rotor in the study encountered flexible support conditions, whereby the stiffness and damping coefficients were determined by the combination of the shaft and bearing components in the rotor system, as shown in Figure 7. The support stiffness consisted of the combination of $K_s$ (shaft stiffness) and $K_b$ (bearing stiffness). It is important to acknowledge that the mass of the shaft, $m_s$, cannot be ignored and it must be lumped into a certain fraction with the mass of the disk, $M_d$. To determine the amount of shaft mass to be lumped with the disk mass, calculations can be carried out using an equivalent system. This entails establishing an equivalent system by analyzing the total kinetic energy (KE) of the shaft under the assumption of vibration in the static deflection mode.

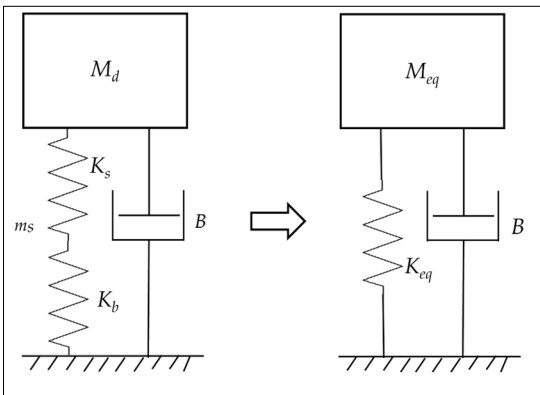

**Figure 7.** Comparison of the Jeffcott rotor to an equivalent 1-D discrete system.

Subsequently, the effective mass of the shaft that should be lumped onto the disk is 0.48 $m_s$. Additionally, the mass of the disk ($M_d$) is increased by 0.48 $m_s$. Therefore, the value of $M_{eq}$ would be 0.96 kg.

The system's overall stiffness is determined by the series connection of the shaft and the bearings, as shown in Figure 7. The stiffness of the shaft under simple supports can be easily calculated to be $K_s = (48EI/L^3)$. Nonetheless, the bearing stiffness $K_b$ is difficult to calculate theoretically; therefore, a practical approach is used to run the rotor experimentally to identify its critical speed ($\omega_{cr}$), which was found to be 2300 rpm for the present case. The identified system critical speed $\omega_{cr} = \sqrt{K_{eq}/M_{eq}}$ can be used in reverse to evaluate the bearing stiffness $K_b$.

The critical speeds associated with X and Y vibrations for an asymmetric rotor are different. However, the asymmetry in the rotor rig is too small to be differentiable in the vibration measurements. While evaluating the critical speeds, it was assumed that the first recorded critical speed pertains to the Y direction. Additionally, it was noted that the critical speed in the X direction, as referenced in prior research [48], is approximately 5% greater than in the Y direction.

The damping ratios in the X and Y directions of the fundamental (1st) mode can be calculated using the logarithmic decrement (LD) from the measured transient response after shutting down the motor from steady operation, as shown in Figure 8. Nevertheless, the LD is designed for a simple 1-dof vibratory system. The shaft-disk-bearing system on the test rig, however, contains an infinite number of modes. The participation of higher modes in the transient response inevitably interferes with the fundamental wave, as seen in the saw-toothed waves. The rotor involves light damping, such that a slight interference at the peak will make the succeeding peak greater than the previous. To avoid the noise from higher frequencies, we picked two adjacent waves with the least interference in the peaks and smoothed them out for the damping calculation. The values obtained after averaging for $\zeta_x$ and $\zeta_y$ were 0.5% and 0.47%, respectively. Table 4 summarizes all model parameters of the rotor-bearing system in the test rig.

The experimental rotor was connected to a computer, enabling the acquisition of data via sensors, which were then analyzed using LabVIEW Version 20.0.1f1 (32-bit) software. The rotor's vibration signals were recorded by the sensors located on the rotor disk, measuring the vibrations in both the X and Y axes. The rotor was subjected to single-frequency excitation and the steady state response, as expected, showed an almost simple harmonic wave, unlike the transient response in Figure 8. Figure 9 demonstrates one example of a recorded sensor response, in which the red-colored dots represent the key phasor pulses in the X and Y directions. The response phase angle $\varphi$ is defined as the angular displacement in degrees from the key phasor pulse to the first positive peak of the vibration amplitude. From Figure 9, the estimated value of $\varphi_x$ is roughly 110°, whereas $\varphi_y$ is approximately 200°, with $\varphi_y$ lagging behind the X probe sensor by 90°. This phase difference somewhat verifies

the correctness of the data sequences, as the Y sensor is precisely positioned 90° behind the X sensor.

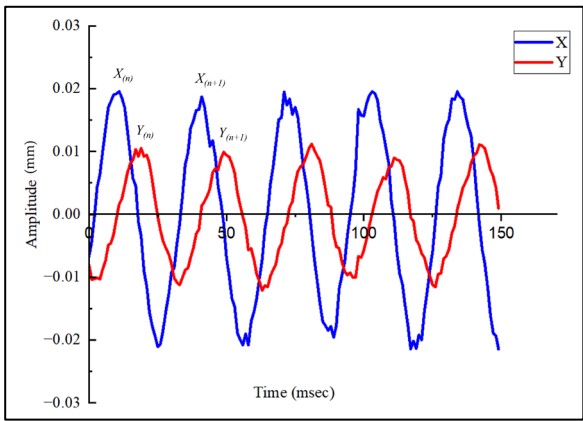

**Figure 8.** Transient responses in the damping evaluation.

**Table 4.** Estimated model parameters of the rotor-bearing system in the test rig.

| C | $M_{eq}$ (kg) | $\zeta$ | $K_s$ (kN/m) | $K_b$ (kN/m) | $K_{eq}$ (kN/m) | $\omega_n$ (rad/s) |
|---|---|---|---|---|---|---|
| Y | 0.96 | 0.47% | 96.908 | 108.92 | 51.282 | 230.1 |
| X | 0.96 | 0.5% | 96.908 | 135.72 | 56.538 | 241.6 |

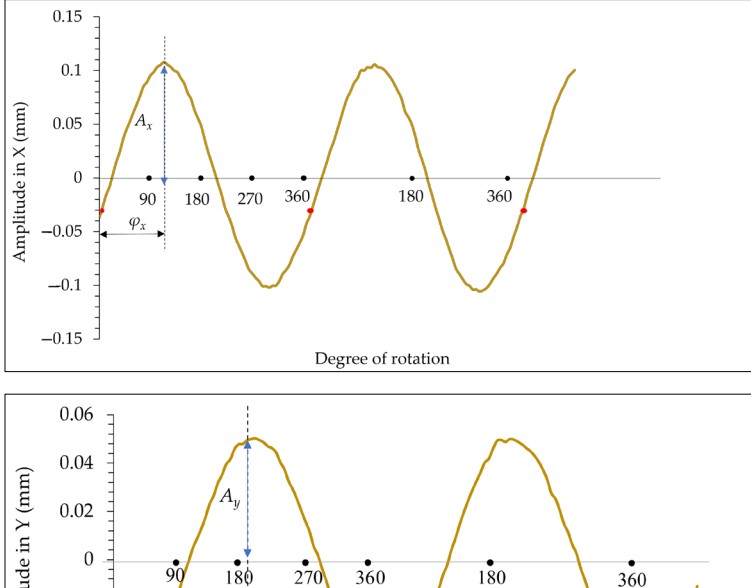

**Figure 9.** Phase angle and amplitude from the measured sensor responses.

The vibration responses from the sensor outputs can be expressed as:

$$u(t) = A_x \cos(\Omega t - \varphi_x) = A_x \cos(\varphi_x) \cos(\Omega t) + A_x \sin(\varphi_x) \sin(\Omega t)$$
$$v(t) = A_y \cos(\Omega t - \varphi_y) = A_y \cos(\varphi_y) \cos(\Omega t) + A_y \sin(\varphi_y) \sin(\Omega t)$$

(27)

where the vibration amplitudes and phases in the X and Y directions can be obtained from Figure 9. The four feature responses ($f_1, f_2, f_3$, and $f_4$) from the experimental study can be assessed using the equations shown above and Equations (20)–(23). This response can be fed into a trained FNN implemented into the monitoring system for real-time fault component ($U, \alpha, s, \theta$) diagnosis.

The disk in the test rig features 16 holes with an angular separation of 22.5° between each hole, facilitating the determination of the imbalance phase angle $\alpha$ measured from the key phasor. The formation of a small permanent shaft bow is difficult to control because the shaft must be heated and bent to reach the plastic deformation region. Therefore, only two shaft bows were employed in the study, each with residual bowing of 0.5 mm and 4 mm, as shown in Figure 10. These shafts were installed on the rotor kit and the mass imbalances on the disk were varied before initiating the rotor operation.

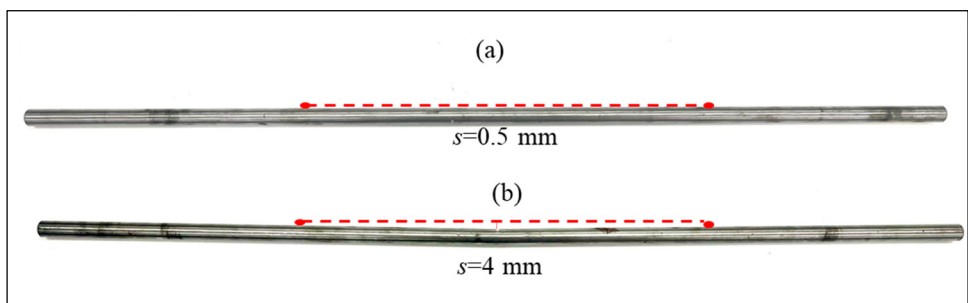

**Figure 10.** (**a**) A shaft bow with $s = 0.5$ mm and (**b**) a shaft bow with $s = 4$ mm.

The operational parameters were deliberately set to 1600 rpm and 3200 rpm, corresponding to the sub-critical ($\tau = 0.69$) and trans-critical speeds ($\tau = 1.39$), respectively, given initial bowing of 0.5 mm. In the interest of safety, the machinery was deliberately operated at a speed far below the critical threshold during the initial bowing phase of 4 mm. This experimental validation approach is used to determine an FNN's capability in identifying fault components related to imbalances and shaft bowing ($U, \alpha, s, \theta$). However, the real fault component was employed with four different values while keeping $s$ constant to estimate the fault components using the FNN. A detailed comparison between the real experimental data and the FNN's estimated values is given in Tables 5–7. Each table illustrates four scenarios of imbalance and shaft bowing combinations—case 1 reveals two fault angles in the same quadrant with a difference, case 2 demonstrates the two fault angles in the phase, case 3 depicts faults almost in the anti-phase, and case 4 demonstrates two faults almost perpendicular to each other, i.e., almost a 90° difference.

In the following tables, the diagnosed errors are expressed as two types. First, they are expressed in terms of the fault amplitude percentage error and the phase difference, as shown in column 5. Columns 6 and 7, however, express the errors in terms of the in-line and vertical amplitude error percentages $\Delta U_{/\!/}$, $\Delta s_{/\!/}$ and $\Delta U_{\perp}$, $\Delta s_{\perp}$. The error in line with the fault vector indicates the amplitude error, while the vertical component can be viewed as the direction error. Based on the results presented in Table 5, it can be observed that under sub-critical speeds and with the amplitude error of concern, the imbalance and shaft bowing errors both exhibit the lowest values, as they are in-phase. The biggest errors for the two faults occur in case 4, in which the two faults are perpendicular to each other. Table 6 shows similar cases to Table 5 except that the rotor was running at a trans-critical speed of 3200 rpm. It can be observed that the lowest error value for the imbalance occurred in case 4 and the lowest bow error value occurred in case 3. A peculiar feature can be observed in Table 6, in which the shaft bowing exhibits significant error values of up to 38% for all cases except case 3, when two faults were in the anti-phase. This is because the bowing response diminishes at high speeds, such that the identification process generates larger errors. Table 7 illustrates the case of $s = 4$ mm at a very slow running speed. Under such conditions, the overall imbalance errors are bigger because the bow strongly dominates the

response, such that the imbalance response becomes a fraction of the total response and generates a larger error during the data fitting process. The lowest imbalance error happens in the anti-phase situation. The overall shaft bow errors are small, except for case 4.

**Table 5.** Diagnosis error percentage at 1600 rpm ($\tau = 0.69$) for initial bowing of 0.5 mm.

| Case | Faults | Real | Diagnosed | Error | Comp | Error% |
|---|---|---|---|---|---|---|
| 1 (same quadrant) | $U$ | 6 | 7.03 | 17.2% | $\Delta U_{//}$ | 17.0 |
| | $\alpha$ | 45° | 48.80° | 3.08° | $\Delta U_{\perp}$ | 6.3 |
| | $s$ | 0.5 | 0.36 | 28.0% | $\Delta s_{//}$ | −28.4 |
| | $\theta$ | 60° | 53.95° | 6.05° | $\Delta s_{\perp}$ | −7.6 |
| 2 (in-phase) | $U$ | 12 | 12.29 | 2.4% | $\Delta U_{//}$ | −0.6 |
| | $\alpha$ | 90° | 76.14° | 13.86° | $\Delta U_{\perp}$ | −24.5 |
| | $s$ | 0.5 | 0.58 | 16.0% | $\Delta s_{//}$ | 14.6 |
| | $\theta$ | 90° | 81.16° | 8.84° | $\Delta s_{\perp}$ | −17.8 |
| 3 (anti-phase) | $U$ | 24 | 21.28 | 11.3% | $\Delta U_{//}$ | −13.1 |
| | $\alpha$ | 225° | 213.63° | 11.37° | $\Delta U_{\perp}$ | −17.5 |
| | $s$ | 0.5 | 0.39 | 22.0% | $\Delta s_{//}$ | −22.3 |
| | $\theta$ | 30° | 35.28° | 5.28° | $\Delta s_{\perp}$ | 7.2 |
| 4 (perpendicular) | $U$ | 30 | 38.24 | 27.5% | $\Delta U_{//}$ | 22.9 |
| | $\alpha$ | 67.5° | 82.88° | 15.38° | $\Delta U_{\perp}$ | 33.8 |
| | $s$ | 0.5 | 0.35 | 30.0% | $\Delta s_{//}$ | −30.0 |
| | $\theta$ | 150° | 147.87° | 2.13° | $\Delta s_{\perp}$ | −2.6 |

**Table 6.** Diagnosis error percentage at 3200 rpm ($\tau = 1.39$) for initial bowing of 0.5 mm.

| Number | Faults | Real | Diagnosed | Error | Comp | Error% |
|---|---|---|---|---|---|---|
| 1 (same quadrant) | $U$ | 6 | 4.4 | 26.7% | $\Delta U_{//}$ | −27.7 |
| | $\alpha$ | 45° | 35.24° | 9.76° | $\Delta U_{\perp}$ | 12.4 |
| | $s$ | 0.5 | 0.69 | 38.0% | $\Delta s_{//}$ | 36.5 |
| | $\theta$ | 60° | 54.41° | 8.59° | $\Delta s_{\perp}$ | 20.6 |
| 2 (in-phase) | $U$ | 12 | 13.34 | 11.2% | $\Delta U_{//}$ | 10.3 |
| | $\alpha$ | 90° | 97.24° | 7.24° | $\Delta U_{\perp}$ | 14.0 |
| | $s$ | 0.5 | 0.31 | 38.0% | $\Delta s_{//}$ | −38.4 |
| | $\theta$ | 90° | 96.18° | 6.18° | $\Delta s_{\perp}$ | 6.7 |
| 3 (anti-phase) | $U$ | 24 | 26.72 | 11.3% | $\Delta U_{//}$ | 2.8 |
| | $\alpha$ | 225° | 202.39° | 22.61° | $\Delta U_{\perp}$ | 42.8 |
| | $s$ | 0.5 | 0.54 | 8.0% | $\Delta s_{//}$ | 5.9 |
| | $\theta$ | 30° | 18.68° | 11.32° | $\Delta s_{\perp}$ | 21.2 |
| 4 (perpendicular) | $U$ | 30 | 28.41 | 5.3% | $\Delta U_{//}$ | −7.0 |
| | $\alpha$ | 67.5° | 56.53° | 10.97° | $\Delta U_{\perp}$ | 18.0 |
| | $s$ | 0.5 | 0.31 | 38.0% | $\Delta s_{//}$ | −39.0 |
| | $\theta$ | 150° | 139.88° | 10.12° | $\Delta s_{\perp}$ | 10.9 |

However, an observable error disparity related to the imbalance and shaft bowing becomes apparent when the rotor is rotating over the critical speed, particularly above 3200 rpm. Investigating the vibration patterns caused by imbalance and shaft bow faults suggests that a smaller discrepancy in inaccuracy occurs for the imbalance when the frequency ratio surpasses one, which can be observed at 3200 rpm in Figure 5. Upon further examination, it is evident that the imbalance error at 3200 rpm exceeds the prediction based on the theoretical explanation. This variation is caused by various practical engineering issues that can impact the results, adding complexities beyond the reduced theoretical models.

For $s$ = 4 mm at 650 rpm in this state, the rotor's frequency ratio is lower than the critical speed. Figure 5 shows that when the frequency ratio is smaller than one, the

system's behavior indicates that the response is mostly influenced by the shaft bowing amplitude. This crucial finding suggests that in specific circumstances, the influence of the shaft bowing on the system's behavior is greater than that of the imbalance. The small discrepancy in the error margins indicates that both types of faults have a substantial role in the observed vibrations, despite their theoretical differences. The consistent error gaps highlight the complex relationship between imbalance and shaft bow effects on the system's dynamic behavior. This shows a situation where both causes have similar effects, and their combined impact is seen in the total inaccuracy in the system's response.

**Table 7.** Diagnosis error percentage at 650 rpm ($\tau$ = 0.28) for initial bowing of 4 mm.

| Number | Faults | Real | Diagnosed | Error | Comp | Error% |
|---|---|---|---|---|---|---|
| 1 (same quadrant) | $U$ | 6 | 5.12 | 14.7% | $\Delta U_{//}$ | −15.9 |
| | $\alpha$ | 45° | 54.71° | 9.71° | $\Delta U_{\perp}$ | 14.4 |
| | $s$ | 4.0 | 4.17 | 4.2% | $\Delta s_{//}$ | 0.8 |
| | $\theta$ | 60° | 74.84° | 14.84° | $\Delta s_{\perp}$ | 26.7 |
| 2 (in-phase) | $U$ | 12 | 14.86 | 23.8% | $\Delta U_{//}$ | 11.7 |
| | $\alpha$ | 90° | 64.39° | 25.61° | $\Delta U_{\perp}$ | 53.5 |
| | $s$ | 4.0 | 4.27 | 6.8% | $\Delta s_{//}$ | −4.7 |
| | $\theta$ | 90° | 116.83° | 26.83° | $\Delta s_{\perp}$ | 48.2 |
| 3 (anti-phase) | $U$ | 24 | 30.28 | 26.0% | $\Delta U_{//}$ | 24.3 |
| | $\alpha$ | 225° | 215.06° | 9.94° | $\Delta U_{\perp}$ | 21.8 |
| | $s$ | 4.0 | 3.72 | 7.0% | $\Delta s_{//}$ | −7.1 |
| | $\theta$ | 30° | 32.71° | 2.71° | $\Delta s_{\perp}$ | 4.4 |
| 4 (perpendicular) | $U$ | 30 | 32.16 | 7.2% | $\Delta U_{//}$ | −5.2 |
| | $\alpha$ | 67.5° | 39.63° | 27.87° | $\Delta U_{\perp}$ | 50.1 |
| | $s$ | 4.0 | 3.0 | 25.0% | $\Delta s_{//}$ | −29.7 |
| | $\theta$ | 150° | 169.19° | 19.9° | $\Delta s_{\perp}$ | 25.4 |

The more significant disparities between the actual and diagnosed faults can be prominently observed for all experiments, unlike the scenarios using simulated data, in which very high accuracy rates were achieved. These larger errors may be attributed to several causes. For instance, there may have been an existing imbalance and bowing before we intentionally added the trial weight or because the notable clearance between the shaft and bearing was not a good fit for the simple support assumption. Both of these issues would generate vibration noises and reduce the diagnosis accuracy in the experiments.

In essence, this research demonstrates the efficacy of the combined physical model and machine learning approach in identifying multiple faults. The integration of the physical model ensures an accurate representation of the Jeffcott rotor system, while the FNN provides efficient and reliable diagnosis capabilities. This research can provide new possibilities for advanced rotor system diagnosis and maintenance strategies in various industrial applications.

## 5. Conclusions

This study proposed a novel hybrid approach that combines model-based and ML approaches to investigate imbalance and shaft bow monitoring for Jeffcott rotor-bearing systems in online settings. Initially, the physical model of the Jeffcott rotor system involved the development of a mathematical model to determine the rotor parameters, generating a substantial amount of simulated fault data for the ANN model. The validated physical model served as the basis for generating a significant volume of simulated data (10,000 datasets), which was subsequently employed to train ML models such as our FNN. This dataset was used in training, validating, and testing the models, ensuring their effectiveness in diagnosing multiple faults. However, the FNN with 40 nodes exhibited the lowest RMSE, indicating better performance. The FNN was tested in diverse conditions, including varying fault and frequency ratios. Under these diverse conditions, using the simulated datasets, the FNN

consistently demonstrated exceptional performance, particularly when the imbalance and shaft bowing did not dominate each other. Moreover, to achieve higher diagnosis accuracy for shaft bowing, the rotor should be run at low speeds, while higher speeds are better for imbalance diagnosis.

The combinations of fault components ($U$, $\alpha$, $s$, $\theta$) were applied to the Jeffcott rotor experiment, and the experimental data were obtained and fed into the trained FNN for instant diagnosis. The FNN effectively identified and quantified faults within the system. The results showed that both the imbalance and bowing faults exhibited their lowest diagnosis error values, as these two faults were in-phase at sub-critical running speeds. At trans-critical speeds, the imbalance error was lowest; however, the bowing error significantly increased because the bowing response rapidly diminished.

These findings highlight the superiority of the hybrid approach and its potential for enhancing the precision and reliability of imbalance and shaft bowing diagnosis in rotor systems. The integration of the physical model ensures an accurate representation of the rotating system, while the machine-learning-based approach provides efficient and reliable diagnosis and monitoring capabilities. This hybrid approach is effective in identifying multiple faults, specifically imbalances and shaft bowing issues, addressing the challenges faced by conventional diagnostic techniques. Consequently, this study provides new possibilities for advanced rotor system monitoring and maintenance strategies in various industrial applications.

**Author Contributions:** Conceptualization and methodology, S.-C.H.; validation, M.N.; formal analysis and software, S.O.; writing—original draft preparation, M.N.; writing—review and editing, S.-C.H. All authors have read and agreed to the published version of the manuscript.

**Funding:** The authors would like to acknowledge the financial support by Taiwan's National Science and Technology Council under the Grant No. NSTC 112-2221-E-131-020.

**Institutional Review Board Statement:** Not applicable.

**Informed Consent Statement:** Not applicable.

**Data Availability Statement:** The original contributions presented in the study are included in the article, further inquiries can be directed to the corresponding authors.

**Conflicts of Interest:** The authors declare no conflicts of interest.

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
