# Peer review of "A Novel Hybrid Approach to the Diagnosis of Simultaneous Imbalance and Shaft Bowing Faults in a Jeffcott Rotor-Bearing System"

_applsci, doi:10.3390/app14083269_

Round 1

Reviewer 1 Report

Comments and Suggestions for Authors

interesting work.

comments:

-it can be useful for the clarity of the text, in my opinion, to separate the introduction and a state-of-the-art paragraph.

- many formulas in paragraph 2 seem not to be in natural proportion and appear as "stretched images" in the longitudinal direction. Please adjust the proportions.

- please introduce the figure in the text before the figure itself.  this comment applies to figure 2 and figure 3.

-row 438 : change "dominating" with "is dominant"

-row 465: "in other words" instead of "in words"

-row 469-470: "diagnosis of the simultaneous..." instead of "diagnoses simultaneous.."

-Table 2: explain what the bold character stands for

-figure 5 : write the measurement unit on y-axis

-row 477: "the bently Nevada experimental rig, model ....... " Write the model of the test rig

-figure 8: put the figure note in the same page as the figure

-rows 526-535 : use justified text, not left aligned text

-figure 9: on the y-axes are missing the measurement units

row 566: put a space after the period

-rows 568-571 . I didn't really understand this part. Two faults means two unbalanced masses? Bow and masses? please explain better, maybe with an additional figure.

-row 588; remove "that"

-table 4: put the table on the same page

- it would be good if you could add a discussion paragraph to discuss your results more. 

-page 9-10. It would be good to add more information about your neural network. How do you initialize it? How long are the computations? 

-moreover, how do you generate the simulated dataset? How many steps do you put in each parameter to generate the combinations? What values do the unbalance parameter and bow parameter reach in the simulations?

Thank you, and good work.

Comments on the Quality of English Language

as above

Reviewer 2 Report

Comments and Suggestions for Authors

The paper proposes an improvement in the early diagnosis of failures in the bearing system of electric motors, this being one of the main mechanical failures that affect the operation and performance of the motor.

The study is novel as it presents for the first time a combined method based on artificial intelligence together with the physical model of the bearings. In this way, the advantages of each method are complemented.

Although the assessment of the review is generally positive, it is suggested to improve in the following aspects:

       Use a specific section for the analysis of the state of the art due to its breadth and depth.

       It is suggested to add some reference to the “bearing current” problem suffered by the motor bearing system caused by variable speed drives.

       The term, imbalance or unbalance, must be unified.

       It is suggested to describe the objective and contribution of this study separately.

Reviewer 3 Report

Comments and Suggestions for Authors

I will present the review by referring to the content of the following chapters. Overall, I evaluate the article positively, however, there are several significant shortcomings that in my opinion must be corrected before publication.

1 Introduction

- I believe it is written inadequately. First of all, it is too general and too broad. The number of publications on the diagnosis of rotating machinery is very large, so I understand the difficulty of the authors. However - review publications are cited, do not indicate specific problems or inadequacies of the proposed work, methods, the solution of which should be the reviewed work. It gets to the point that the authors in the introduction describe what unbalance is, citing the paper [31]. This is engineering knowledge from the early years of study - it is really unnecessary. One could conclude that the entire introduction consists of abstracts, or fragments thereof, of randomly referenced papers. It is absolutely necessary to rewrite the introduction choosing the topic of the article as the focus, problems that other authors have encountered when trying to solve similar problems. 

- In addition, you should avoid announcing sentences that add nothing to the understanding of the essence of the work - see, for example, lines 54-56.

Chapter 2

The entire chapter is understandable and correctly described with some minor shortcomings, noted below:

- Figure 1 should be improved. There is a lot of space in the text, a plane could be added to better visualize the marked angles, and the labels could be enlarged for better readability.

- What is meant by "possible damping" B - this needs to be clarified.

- ":" should be added in lines 242, 274, 

- The statement in lines 298-301 that damping has little effect on the response of the system is not obvious to me, it seems reasonable to prove this statement.

Chapter 3

Written understandably. Unfortunately, in my opinion, the novelty of the proposed approach has not been sufficiently proved. The authors themselves point in line 59 to the hybrid approach as one of the approaches used. What, then, is the novelty of their approach compared to others already in operation in industrial or laboratory practice, if only?

- Figure 2 should be revised to improve readability

4 Chapter 4

Critical comments:

- Figure 8 shows the time waveform from which the damping was estimated. It is very doubtful that the result is correct, please look even at the last maximum of the waveform - the damping will be negative, the amplitude is increasing after all. Please make a check or choose and present another method of estimating attenuation.

- I do not know for what purpose in the text are Figures 10 and 11. Both do not contribute anything to understanding the content of the article. Please remove them.

- Missing space in line 566

Round 2

Reviewer 3 Report

Comments and Suggestions for Authors

Changes were made in accordance with comments.